# Self-capping of nucleoprotein filaments protects the Newcastle disease virus genome

Xiyong Song[1,2,7†], Hong Shan[3†], Yanping Zhu[2†], Shunlin Hu[4], Ling Xue[4], Yong Chen[2,7], Wei Ding[6], Tongxin Niu[6], Jian Gu[4], Songying Ouyang[5,8*], Qing-Tao Shen[3*], Zhi-Jie Liu[1,2,3*]

[1]Institute of Molecular and Clinical Medicine, Kunming Medical University, Kunming, China; [2]National Laboratory of Biomacromolecules, Institute of Biophysics, Chinese Academy of Sciences, Beijing, China; [3]iHuman Institute, ShanghaiTech University, Shanghai, China; [4]College of Veterinary Medicine, Yangzhou University, Yangzhou, China; [5]The Key Laboratory of Innate Immune Biology of Fujian Province, Biomedical Research Center of South China, Key Laboratory of OptoElectronic Science and Technology for Medicine of Ministry of Education, College of Life Sciences, Fujian Normal University, Fuzhou, China; [6]Center for Biological Imaging, Institute of Biophysics, Chinese Academy of Sciences, Beijing, China; [7]College of Life Sciences, University of Chinese Academy of Sciences, Beijing, China; [8]Provincial University Key Laboratory of Cellular Stress Response and Metabolic Regulation, College of Life Sciences, Fujian Normal University, Fuzhou, China

*For correspondence:
ouyangsy@fjnu.edu.cn (SO);
shenqt@shanghaitech.edu.cn (Q-TS);
liuzhj@shanghaiTech.edu.cn (Z-JL)

†These authors contributed equally to this work

Competing interests: The authors declare that no competing interests exist.

**Abstract** Non-segmented negative-strand RNA viruses, such as measles, ebola and Newcastle disease viruses (NDV), encapsidate viral genomic RNAs into helical nucleocapsids, which serve as the template for viral replication and transcription. Here, the clam-shaped nucleocapsid structure, where the NDV viral genome is sequestered, was determined at 4.8 Å resolution by cryo-electron microscopy. The clam-shaped structure is composed of two single-turn spirals packed in a back-to-back mode. This tightly packed structure functions as a seed for the assembly of a nucleocapsid from both directions, facilitating the growth of double-headed filaments with two separate RNA strings inside. Disruption of this structure by mutations in its loop interface yielded a single-headed unfunctional filament.

DOI: https://doi.org/10.7554/eLife.45057.001

## Introduction

Members of the order *Mononegavirales* encompass some of the most lethal human and animal pathogens, including ebola, rabies virus, measles, nipah virus and the human respiratory syncytial virus (RSV) (*Amarasinghe et al., 2017*; *Kuhn et al., 2010*). *Mononegaviruses* commonly contain a non-segmented, linear, negative-strand RNA genome, and the replication of this genome is vital for virus survival and pathogenicity (*Ruigrok et al., 2011*). One remarkable character of negative-strand RNA viruses is that their genomes are enwrapped by the nucleoprotein (N), which results in the formation of helical nucleocapsids (*Finch and Gibbs, 1970*; *Heggeness et al., 1980*; *Longhi, 2009*). During viral RNA synthesis, the assembled nucleocapsid, rather than the naked RNA genome, is opened and unveiled so that it can be recognized by the viral RNA-dependent RNA polymerase (RdRp) and it serves as the template for both replication and transcription (*Dochow et al., 2012*; *Emerson and Wagner, 1972*; *Emerson and Yu, 1975*; *Fearns et al., 1997*; *Perlman and Huang,*

*1973*; *Severin et al., 2016*). In *Paramyxoviridae* or *Rhabdoviridae* viruses, the viral phosphoprotein (P) mediates the ability of RdRp to access nucleoprotein, and the RdRp moves across the nucleocapsid for viral transcription (*Blanchard et al., 2004*; *Bourhis et al., 2006*; *Kingston et al., 2004*). RNA is susceptible to nuclease in vivo, so the virus has evolved a complicated mechanism to protect its viral genome, in which its N plays a major role in enwrapping nascent RNA thereby preventing possible damage (*Dortmans et al., 2010*; *Ruigrok et al., 2011*).

A great deal of effort has been expended on understanding the N assembly mechanism that protects the genome of the *Mononegavirales.* N has two domains, the amino-terminal domain (NTD) and the carboxy-terminal domain (CTD), with a positively charged cleft in between that is suitable for RNA binding. In the presence of RNA, each N can bind 6, 7 or 9 nucleotides and thus can clamp RNA into the cleft, forming a ribonucleoprotein complex (RNP) (*Albertini et al., 2006*; *Gutsche et al., 2015*; *Tawar et al., 2009*). RNP can further assemble into either a helical or ring structure with 10, 11 or 13 protomers per turn (*Albertini et al., 2006*; *Green et al., 2006*; *Gutsche et al., 2015*; *Tawar et al., 2009*). In RNP oligomers, the NTD and CTD interact successively with adjacent N proteins, forming long helical filaments that efficiently protect the viral genome, and which serve as the template for viral RNA transcription and the replication of new virions (*Ge et al., 2010*; *Zhou et al., 2013*).

Detailed structural analyses have shown that measles RNP filaments exhibit more rigid and regular single-headed, herringbone-like characteristics after trypsin treatment, and in this state are seemingly not sufficient to protect RNA genome at the tips of the filaments (*Schoehn et al., 2004*). The mechanism through which viral RNP protects its tips from digestion by proteases remains to be discovered. Here, the Newcastle disease virus (NDV), a member of the genus *Avulavirus*, family *Paramyxoviridae*, which is relatively safe for handling, was selected as the model to look at how NDV RNP protects its viral genome and to provide new insights into the development of nucleocapsid-based antivirus therapies.

## Results

### Clam-shaped NDV nucleocapsid

Following previous reports (*Guryanov et al., 2015*; *Peng et al., 2016*), the NDV N was expressed in an *Escherichia coli* system and pure protein was obtained after tandem affinity and gel-filtration chromatography. The N was found to be of high purity in SDS-PAGE, with an absorbance of A260/A280 of ~1.1, suggesting the presence of RNA-bound N (*Figure 1—figure supplement 1A*). Under negative-stain EM, purified N exhibited round-shaped structures with a small portion of double-headed filaments of different lengths (*Figure 1—figure supplement 1B*), which are similar to the nucleocapsids of measles that are expressed in Sf21 insect cells (*Jensen et al., 2011*), of sendai virus in mammalian cells (*Buchholz et al., 1993*) and of hendra virus in *E. coli* (*Communie et al., 2013*). Those two kinds of assemblies were further separated with continuous sucrose-gradient ultracentrifugation. The separated round-shaped sample was quite homogenous with a diameter of ~200 Å and was used for subsequent structure determination (*Figure 1A*).

The cryo-electron microscopy (cryo-EM) images for the round-shaped samples were collected and the single particle analysis was carried out. Two-dimensional (2D) and three-dimensional (3D) classification results showed a clam-shaped rigid body with some flexible extensions (*Figure 1—figure supplement 1C–E*). Further 3D refinement resolved the clam-shaped structure to 6.4 Å resolution and showed obvious C2 symmetry in the rigid body. The C2 symmetry was then applied to improve the resolution, yielding an overall 4.8 Å resolution of the core structure (*Figure 1B*, *Figure 1—figure supplement 2A–E* and *Video 1*). Each protomer was easily recognized from the reconstruction. Those protomers furthest from the seam were better resolved while those closer to the seam were of lower resolution (*Figure 1—figure supplement 2E*). However, an atomic resolution structure of NDV N was still missing. Homolog modeling on NDV N, based on the 40% sequence identity of N between NDV and Parainfluenza virus 5 (PIV5) (*Alayyoubi et al., 2015*), resulted the subunit N model and the model was flexibly docked into the EM density map (*Figure 1C*, *Figure 1—figure supplement 3A,B*). The docked model fits the EM density well with only minor modification, and resulted in a reliable initial model of NDV N.

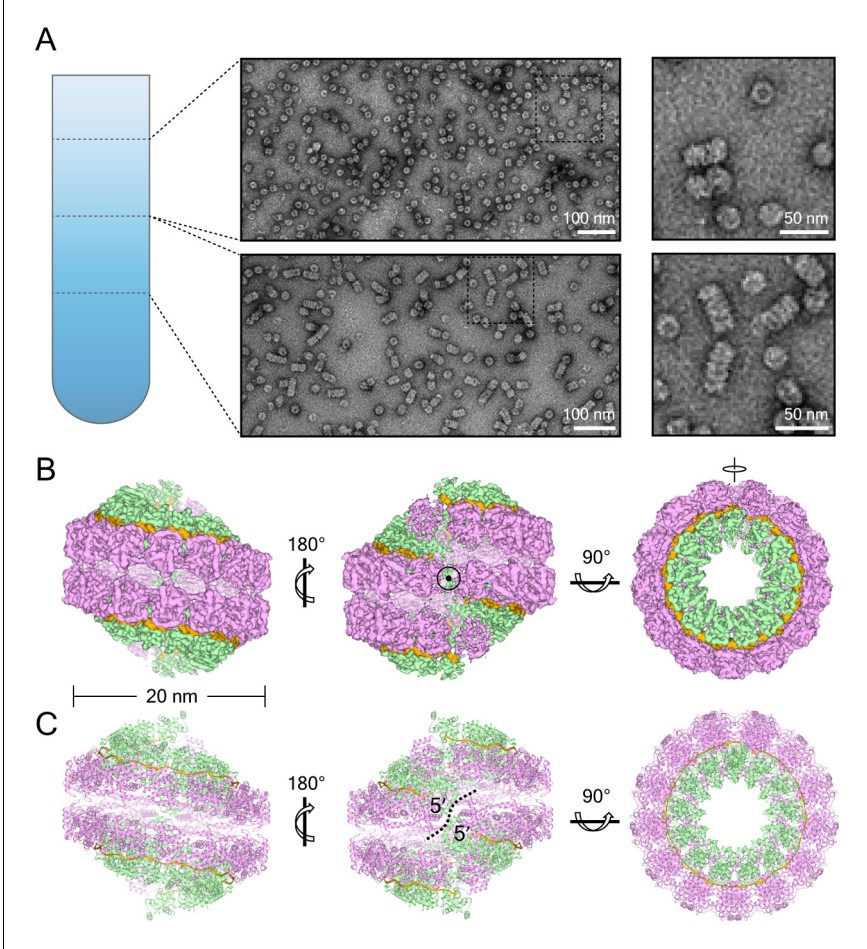

**Figure 1.** NDV N assembles into clam-shaped structures with two single-strand spirals packing in a back-to-back manner. (**A**) The images show negative-stain EM micrographs of the round-shaped structures (top image, upper fraction) and filaments (bottom image, lower fraction) after sucrose-gradient centrifugation (close-ups of the boxed areas are shown on the right). (**B**) Various views of the 3D reconstruction of the clam-shaped structure of N from the upper fraction. The C2 symmetry axis enforced during reconstruction is indicated in the center view (middle). The NTD, CTD and RNA are colored in pink, green and gold, respectively. (**C**) Atomic model of the clam-shaped structure of N shown from the same view as in (**B**) and using the same color code. The two 5′ ends of the enwrapped RNA and the seam between them are labeled in the middle view.

DOI: https://doi.org/10.7554/eLife.45057.002

The following figure supplements are available for figure 1:

**Figure supplement 1.** Data analysis of the clam-shaped structure of NDV N.
DOI: https://doi.org/10.7554/eLife.45057.003

**Figure supplement 2.** 3D reconstruction of the clam-shaped structure of N and resolution estimations.
DOI: https://doi.org/10.7554/eLife.45057.004

**Figure supplement 3.** Structural analysis of the clam-shaped structure of NDV N.
DOI: https://doi.org/10.7554/eLife.45057.005

**Figure supplement 4.** The negative-stain EM of the N–RNA complex isolated from NDV.
DOI: https://doi.org/10.7554/eLife.45057.006

The whole reconstruction revealed a clam-shaped structure with the symmetry axis perpendicular to the spiral axis, where two single-turn spirals pack in a back-to-back manner (*Figure 1B,C*). In each single-turn spiral, there are around 13 N molecules per turn, and each N uses its N-arm (residues 2–34) and its C-arm (residues 370–398) to interact horizontally with a neighboring N for domain exchange contact (*Figure 1C* and *Figure 1—figure supplement 3C and E*), as reported in previous ring structures (*Alayyoubi et al., 2015*; *Albertini et al., 2006*; *Green et al., 2006*; *Tawar et al.,*

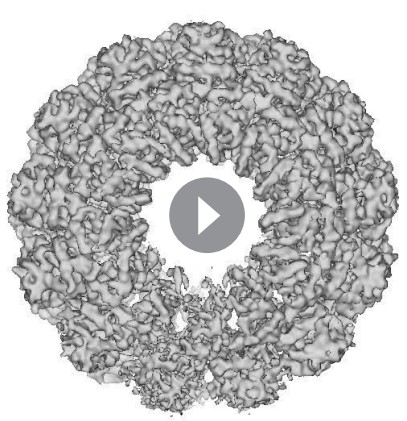

**Video 1.** 3D reconstruction of the clam-shaped structure and the fitting of a pseudoatomic model.
DOI: https://doi.org/10.7554/eLife.45057.007

*2009*). More specifically, in a NDV clam-shaped structure, $N_i$ uses the N-arm to interact with the $N_{i-1}$ CTD and the C-arm to make extensive contact with the $N_{i+1}$ CTD tip, forming an exceedingly stable structure (*Figure 1—figure supplement 3C–E*). Different from the ring structure of PIV5 N (*Alayyoubi et al., 2015*), the add-on N shifts upward by ~4.6 Å, which drives NDV N to form a single-turn spiral instead (*Video 2*).

Endogenous RNA from *E. coli* can be traced in the EM map. Limited by resolution, poly-Uracil (poly-U) was modeled into the EM map to mimic cellular RNA. In the clam-shaped structure, the RNA follows a relaxed helical pattern and orients outside the N molecule, being more similar to the RSV nucleocapsid than to that of rhabdovirus or vesicular stomatitis virus (*Figure 1—figure supplement 3E*, *Video 3*) (*Albertini et al., 2006*; *Green et al., 2006*; *Tawar et al., 2009*). The external RNA is deeply buried in the interdomain cleft between the NTD and the CTD, following

the 'rule of six' with alternating three-base-in and three-base out conformation (*Figure 1—figure supplement 3F*) (*Calain and Roux, 1993*; *Kolakofsky et al., 1998*). Six nucleotides are covered by one N, and there will be 78 nucleotides per single-turn spiral (*Figure 1—figure supplement 3F*). On the basis of nucleocapsid structural similarity between NDV and the measles virus, the RNA in NDV is estimated to be left-handed with the 5' end of RNA, which would be first replicated and enwrapped by N immediately after synthesis, lying inside (as labeled in *Figure 1B and C*) (*Gutsche et al., 2015*).

Of particular note is an obvious seam between the two single-turn spirals, which disconnects two RNA molecules (*Figure 1B,C*). The separation between the two 5' ends of the RNAs is ~6 nm and the bending angle of these ends is approximately 120°, which blocks the continuity of the RNA because it is impossible for one RNA to span two back-to-back spirals. Thus, the clam-shaped structure is not an integrated helix at all, but rather is composed of two spirals self-capping each other in a back-to-back mode. To confirm whether the NDV nucleocapsid is packed using this mode in vivo, the negative-stain EM images of highly polymeric RNPs extracted from Newcastle disease virus were obtained. Interestingly, the images showed a filamentous assembly of the genomic

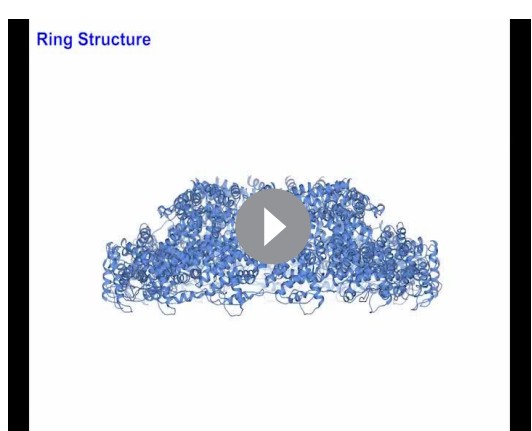

**Video 2.** Morphing of the ring structure to form a single-turn spiral.
DOI: https://doi.org/10.7554/eLife.45057.008

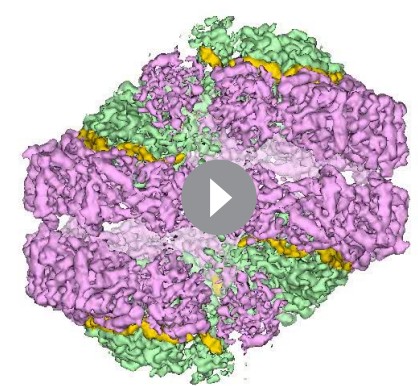

**Video 3.** RNA enwrapped between the NTD and the CTD.
DOI: https://doi.org/10.7554/eLife.45057.009

RNA with the clam-like structure, similar to that observed in the resolved structure (*Figure 1—figure supplement 4*).

## Double-headed filament derived from clam-shaped nucleocapsid

Importantly, decreasing threshold values to show more EM densities with or without C2 symmetry revealed that each single-turn spiral had the potential to grow further into a longer filament following a helical trajectory (*Figure 1—figure supplement 2A,B*). The pseudo-model of N could be docked into extra densities following the helical trajectory without any structural conflicts. Iteratively adding N protein in such manner to both single-turn spirals would yield a longer helix with double heads (*Video 4*).

To verify this, the filaments fraction after ultracentrifugation was examined with cryo-EM. Almost every filament had double heads derived from one clam-shaped structure (*Figure 2A*). Owing to the heterogeneity of the double-headed filaments, their structures could not be directly resolved via the single particle reconstruction approach. Consequently, the filament was split into two parts for structural analysis: the clam-shaped core and the helical part (*Figure 2A,B*). For the former, 2608 clam-shaped particles truncated from double-headed filaments yielded a 14.0 Å resolution structure (*Figure 2—figure supplement 1A*). The overall shape of the core fitted very well with the 4.8 Å clam-shaped structure (*Figure 2B* and *Figure 2—figure supplement 1B*). The distinctive back-to-back packing mode and the seam between two single-turn spirals were clearly recognizable, suggesting that the clam-shaped core acts as the seed for filament growth (*Figure 2D*).

The helical part of the filament was reconstructed at 15.0 Å resolution (*Figure 2B* and *Figure 2—figure supplement 1A*). Like the clam-shaped structure, the helix was composed of 13 protomers per turn, with an outer diameter of ~200 Å, in agreement with the pseudo-atomic model. The helical pitch varied by ~60 Å, which provided flexibility for the helical nucleocapsids to fit into the crowded virus. Thus, the clam-shaped structure was perfectly compatible with the helical filament and could further grow into a helical filament (*Figure 2C,D*). Following the direction of RNAs in the clam-shaped structure, the 5' ends of the RNAs of the double-headed filament were depicted similar to those of the clam-shaped structure (*Figure 1C* and *Figure 2C,D*).

Interestingly, the lengths of the two helixes in around 90% of the back-to-back spirals were not equal and one helix was obviously longer than the other one in the raw images (*Figure 2A* and *Figure 2—figure supplement 1C*). The statistics showed that the shorter helix had an average length of ~14 nm with fewer than two helical turns, while the average length of the longer one was doubled to ~34 nm (*Figure 2E*), although the factors that determine the length difference are uncertain.

## The clam-shaped nucleocapsid affects the function of the viral genome

In the clam-shaped structure or in the derived double-headed filament, the self-capping interface came from loops (residues 114–120) of vertically adjacent N in the clam-shaped core. Distance analysis of the residues in the loop suggested that hydrogen bonds may exist between two pairs of $Glysin_{119}$ and $Arginine_{117}$ residues (*Figure 3A,B*). The $Loop_{114–120}$ region was involved only in the assembly of the clam-shaped core but not in the helical assembly of the double-headed filament. All of the residues in $Loop_{114–120}$ were mutated to Alanine to check whether the mutations affected the clam-shaped assembly. The mutated N ($N_{Loop}$) was purified using the same protocol as that used for $N_{WT}$ and yielded filaments that were an average of 50 nm longer than those of $N_{WT}$. A zoomed-in view of the $N_{Loop}$ filaments clearly showed a single-headed, herringbone-like filament instead of a double-headed assembly from 2D classification (*Figure 3C* and *Figure 2—figure supplement*

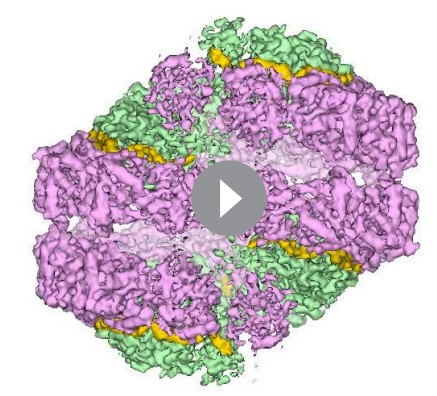

**Video 4.** Double-headed spiral derived from clam-shaped structure.
DOI: https://doi.org/10.7554/eLife.45057.010

2). Direct fast fourier transformation (FFT) analysis of one single-headed $N_{Loop}$ filament showed clean diffraction bands with ~1/60 Å intervals, and the 3D reconstruction of $N_{Loop}$ showed a helical structure that was similar to the single-turn spiral of the double-headed $N_{WT}$. The evidence suggests that $Loop_{114–120}$ has no influence on helical assembly but has a crucial role in clam-shaped structure formation (*Figure 2—figure supplement 2*).

To further investigate whether the $Loop_{114–120}$ is functionally relevant in vivo, minigenome analyses of several N mutants (*Figure 3D,E* and *Figure 3—figure supplement 1*) were performed. The N-arm and the C-arm had been proven previously to be critical for the assembly of N (*Buchholz et al., 1993*; *Kho et al., 2003*) and the truncations of $N_{ΔN-arm}$, $N_{ΔN-armΔC-armΔC-tail}$ and $N_{ΔC-armΔC-tail}$ disabled or heavily affected the assembly to higher ordered structure that was enabled by $N_{WT}$, $N_{Loop}$ and $N_{ΔC-tail}$, as shown by size-exclusion chromatography and negative-stain EM images (*Figure 3D,E* and *Figure 3—figure supplement 1*). RNA synthesis was fully functional in the presence of wild type N ($N_{WT}$), but truncation mutants lacking the N-arm ($N_{ΔN-arm}$), the N-arm or the C-arm/C-tail ($N_{ΔC-armΔC-tail}$ and $N_{ΔN-armΔC-armΔC-tail}$) were all nonfunctional and lost the ability to express the GFP reporter. Surprisingly, the $N_{ΔC-tail}$ were partially functional according to the weak fluorescence signals observed in the minigenome assay. Although the mutation of $N_{Loop}$ could form longer single-headed filaments as mentioned above, it showed a negative result in the fluorescence assay (*Figure 3D,E* and *Figure 3—figure supplement 1* ). RNA replication, transcription or translation was not successful in the minigenome assay of the $N_{Loop}$, indicating that the clam-shaped structure is critical for the expression of the GFP reporter gene.

## The clam-shaped nucleocapsid is resistant to nuclease

The detailed structural analysis showed that the single-headed filament from $N_{Loop}$ exposed the RNP 5′ end to the environment, whereas the double-headed filament from $N_{WT}$ enabled intermolecular self-capping to cover its sensitive 5′ end. (*Figure 4A,B*). To test the sensitivity of the RNP 5′ end to protease, elastase was incubated with double-headed or single-headed filament samples. The SDS-PAGE gel showed a ~40-kDa band with some smears from elastase-digested $N_{WT}$ (*Figure 4C*). Peptide mapping of the 40-kDa band via Mass Spectrum showed a residue range of 33 to 361, which suggested that the cleaving site was in the loop in the C-arm (*Figure 4D* and *Figure 4—figure supplement 1A*). For the single-headed filament from $N_{Loop}$ with the 5′ end exposed, an obvious difference was that the 40-kDa band was found to be further digested to 30 kDa from the N-arm after elastase treatment, based on the SDS-PAGE and Mass Spectrum results, which strongly indicates that there is another cleavage site in the NTD loop regions.

In addition, the influence of nuclease on RNA genome stability was tested. RNase A was added to the solutions containing $N_{WT}$ or $N_{Loop}$ filaments to check the digestion result of the assemblies. The results showed that $N_{Loop}$ was more sensitive to RNase A than that of $N_{WT}$ after 180 s exposure (*Figure 4—figure supplement 1B*). The statistical results showed that almost all of the $N_{Loop}$ samples were completely disassembled whereas over 25% of the $N_{WT}$ filaments remained intact (*Figure 4E*). Meanwhile, the $N_{WT}$ rather than the $N_{Loop}$ contained RNA with an absorbance of A260/A280 of ~0.9, whereas that of $N_{Loop}$ was ~0.6. The enzyme digestion analysis showed that $N_{WT}$, rather than $N_{Loop}$, was resistant to the digestion of nuclease and protease, from which it could be hypothesized that the $N_{Loop}$ exposed its RNP 5′ end without the self-capping protection, and was exposed to protease and became accessible by nuclease. Through self-capping, N can protect the viral genome not only from side attack but also from both ends.

## Discussion

N is the key factor for protecting the nascent RNA from degradation during RNA replication. Different from the reported ring-structure and helical spirals (*Alayyoubi et al., 2015*; *Albertini et al., 2006*; *Green et al., 2006*; *Tawar et al., 2009*; *Gutsche et al., 2015*), a novel clam-shaped structure of NDV N with two single-turn spirals packing in a back-to-back manner was identified and determined, corresponding to the extracted nucleocapsid assembly of NDV (*Figure 1—figure supplement 4*). The clam-shaped structure of the NDV nucleocapsid was verified by in vivo transcription and translation experiments with minigenome analysis. The deletion of the N-arm or the C-arm of N disrupted or affected the formation of highly ordered nucleocapsid and resulted the absence of fluorescence signals in the minigenome assay. However, in a similar minigenome assay in which a

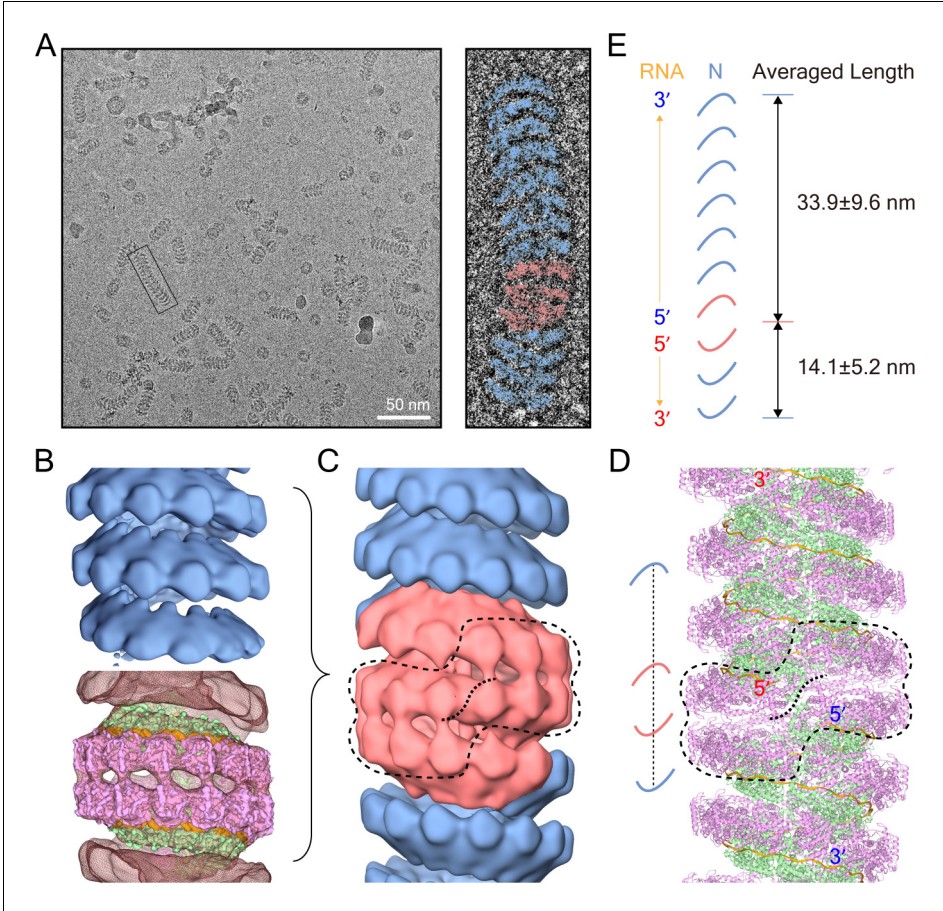

**Figure 2.** Double-headed filament derived from the clam-shaped structure. (**A**) Representative cryo-EM micrograph of the N filament from the lower fraction. One typical double-headed filament was selected, magnified and colored in blue (helical structure) and red (clam-shaped core). (**B**) 3D reconstructions of the helical structure (top) and the clam-shaped core (bottom). The 4.8 Å clam-shaped structure is docked into the clam-shaped core in the filament. (**C**) Combination of both helical filaments and the clam-shaped core yields the whole double-headed filament. The position of the clam-shaped core in the composite structure is delineated by the dashed line. (**D**) Atomic model of the double-headed filament shows the position of the clam-shaped core (dashed line). Corresponding 5′ and 3′ ends from the same RNA are labeled in red and blue. (**E**) The two helixes in one double-headed filament are of unequal length. The length of each helix is defined as the distance between the helix tip and the center of the clam-shaped core in the cartoon. The length measurements and the RNA direction from 5′ to 3′ are given.

DOI: https://doi.org/10.7554/eLife.45057.011

The following figure supplements are available for figure 2:

**Figure supplement 1.** Structural analysis of the double-headed filament.
DOI: https://doi.org/10.7554/eLife.45057.012
**Figure supplement 2.** Structural analysis of a single-headed filament of the $N_{Loop}$.
DOI: https://doi.org/10.7554/eLife.45057.013

---

truncated C-tail, namely $N_{\Delta C\text{-}tail}$, was used, some weaker fluorescence signals were observed. The previous studies showed that the P protein used its NTD domain ($P_{NTD}$) to uncoil the nucleocapsid and allowed the RdRp to access the genomic RNA and then tethered the RdRp to the nucleocapsid with its XD domain binding to the C-tail of the N protein (*Cox et al., 2014*). One possible explanation is that even though the $N_{\Delta C\text{-}tail}$ lacks the C-tail, the P protein can still mediate the formation of the N–RNA–RdRp complex in the minigenome assay due to the interaction of $P_{NTD}$ with the nucleocapsid.

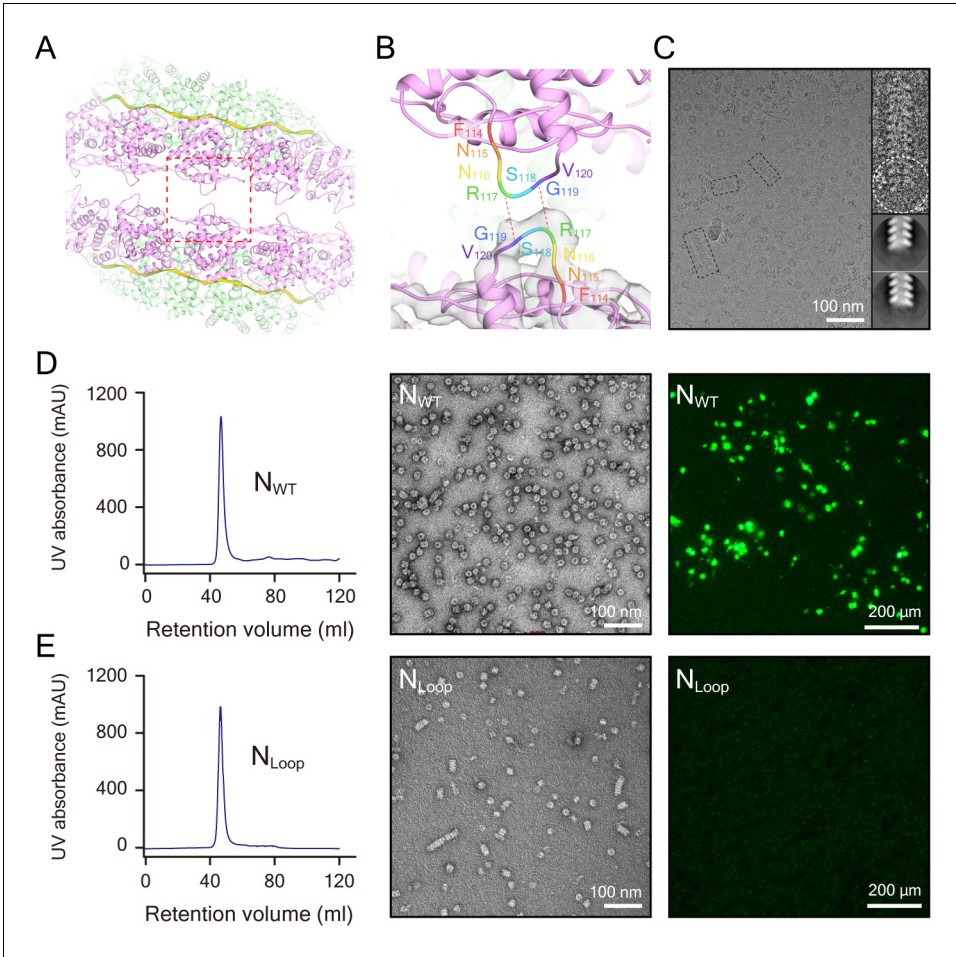

**Figure 3.** The clam-shaped nucleocapsid is important for the function of the viral genome. (**A**) Loop pairs from the vertically adjacent N form the self-capping interface in the clam-shaped structure. Five loop pairs furthest from the seam are shown. Colors are as in *Figure 1*. (**B**) View of one loop pair of the clam-shaped structure. Seven residues (114–120) in the upper loop are labeled and the lower loop is docked into the EM density. (**C**) Raw micrograph of a single-headed helix from the $N_{Loop}$ and the 2D classification of the filament tip (circled). Zoomed-in view of selected raw filaments (examples in dashed boxes) with two typical 2D classes on the tip shown. (**D**) $N_{WT}$ was able to form double-headed filaments and functioned well in the minigenome assay. The retention volume of $N_{WT}$ in gel filtration chromatography was ~47 ml (left) and the negative-stain image of this fraction consisted of a clam-shaped structure and filaments was zoomed in (middle). $N_{WT}$ exhibited strong fluorescence signals in a minigenome assay in BSR-T7/5 cells (right). (**E**) The $N_{Loop}$ formed filaments but was not functional in a minigenome assay. The retention volume of the $N_{Loop}$ was ~47 ml, close to $N_{WT}$ (left). Negative-stain EM showed more filaments than $N_{WT}$ (middle). However, there was no fluorescence signal in the minigenome assay (right).
DOI: https://doi.org/10.7554/eLife.45057.014

The following figure supplement is available for figure 3:

**Figure supplement 1.** The summary and comparison of N and the derived mutants in the nucleocapsid assembly and their function in a minigenome assay.
DOI: https://doi.org/10.7554/eLife.45057.015

---

The double-headed spiral uses a self-capping mechanism that involves the clam-shaped core to protect the RNA genome's integrity. This is the possible reason why the single-headed $N_{Loop}$ was not functional in a minigenome assay. More interestingly, this clam-shaped structure functions as a seed for the assembly of double-headed spirals with two separate RNAs inside. The illumination of the clam-like nucleocapsid expands our understanding of the involvement of N in the assembly of the the helical nucleocapsid, especially introducing the clam-like core as the starting point for N assembly and then elongation on both sides, which maintains genome integrity in vivo. A self-

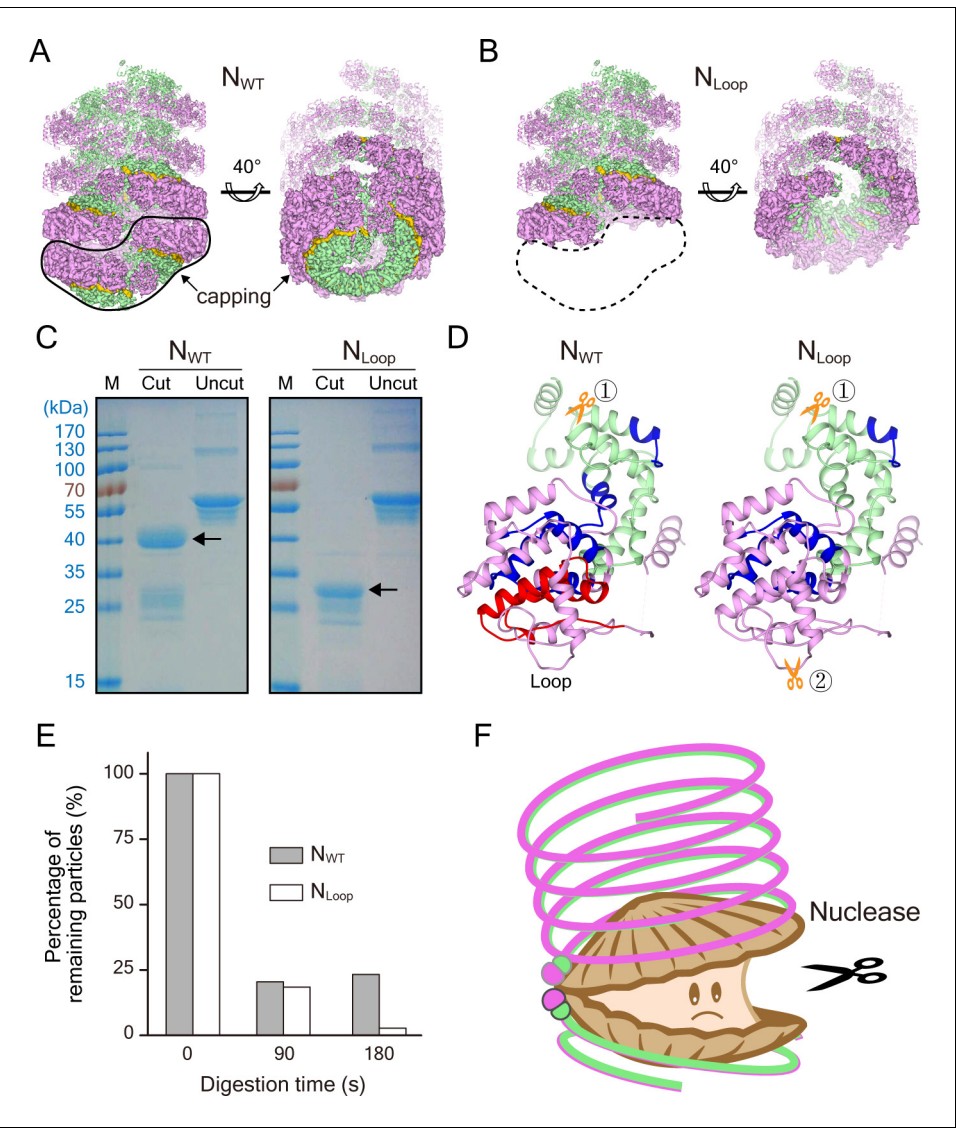

**Figure 4.** Clam-shaped nucleocapsid is resistant to elastase and RNase A. (**A**) An atomic model of an $N_{WT}$ double-headed filament from different views shows reciprocal capping between two single-headed spirals. Colors as in *Figure 1*. One single-headed spiral is highlighted by the black line and labeled 'capping'. (**B**) An atomic model of an $N_{Loop}$ single-headed filament from different views with no cap and with the 5' end of its RNA exposed. The supposed capping spiral, marked by the dashed line, is missing from the single-headed filament. (**C**) SDS-PAGE gels of $N_{WT}$ and $N_{Loop}$ after elastase digestion. There was a ~40-kDa main band with some smears in the elastase-digested $N_{WT}$ assay (left), while elastase cut the $N_{Loop}$ to form a ~30-kDa band (right). (**D**) Mass spectrum results identified peptides drawn on the atomic structure of N, indicating one additional cutting site on $N_{Loop}$ (gold scissors). The common regions mapped by Mass Spectrum in $N_{WT}$ and $N_{Loop}$ are colored in blue, and the unique region checked in $N_{WT}$ is shown in red. Five peptides were identified from the 40-kDa band of $N_{WT}$ and marked on the N atomic structure, leaving the CTD loops as the cutting site. Only four peptides were identified from the 30-kDa band of $N_{Loop}$ and marked on N atomic structure. Given the reduced molecular weight and the missing NTD peptide, another cutting site should exist within NTD. (**E**) Comparison of RNase A digesting $N_{WT}$ and $N_{Loop}$ at different timepoints. Both the clam-shaped structures and the filaments are counted. The numbers of oligomers in $N_{WT}$ and $N_{Loop}$ at 0 s are normalized to 100%. At 180 s, almost 100% disassembly of nucleocapsid was seen in $N_{Loop}$ whereas over 25% of filaments remained in $N_{WT}$. (**F**) A cartoon depicts the hypothetical full protection provided to the viral RNA genome by $N_{WT}$ via the self-capping clam-shaped structure. When the clam-shaped structure is broken, nuclease is able to access the RNA 5' end and can digest the whole RNA strand.
DOI: https://doi.org/10.7554/eLife.45057.016

The following figure supplements are available for figure 4:

*Figure 4 continued on next page*

*Figure 4 continued*

**Figure supplement 1.** Peptide mapping and RNase A digestion of $N_{WT}$ and $N_{Loop}$.
DOI: https://doi.org/10.7554/eLife.45057.017
**Figure supplement 2.** C-tail may be located inside the clam-shaped structure.
DOI: https://doi.org/10.7554/eLife.45057.018

capping mechanism is quite common in filaments that are involved in biological processes. For example, in DNA repair, the Rad51 paralog complex RFS-1/RIP-1 induces remodeling at the tips of Rad51–ssDNA filaments to stimulate Rad51 strand exchange activity (*Taylor et al., 2015*; *Taylor et al., 2016*). In microtubule assembly, γ-tubulin pre-assembles into single-turn spirals that serve as the template to nucleate sequential α/β-tubulin assembly (*Kollman et al., 2010*; *Zehr et al., 2014*). Previous studies have shown that self-capping is a mechanism that allows proteins or their homologues to fit easily into the spiral assembly and to fine-tune its function efficiently.

The clam-shaped nucleocapsid also provides possible explanations for the pleomorphism and polyploidy of the *Mononegavirales*. Mononegavirus morphology appears to vary considerably, especially among the *Paramyxoviruses* and *Filoviruses*, in the range of about 110–540 nm in diameter for Sendai virus (*Loney et al., 2009*), 100–250 nm for spherical NDV particles (*Battisti et al., 2012*), and 50–510 nm for the measles virus (MeV) (*Cox and Plemper, 2017*). This character of flexible volume of virus could accommodate variation in the copy number of the genome. It is a common observation for mononegaviruses to contain more than one genome; examples include NDV (*Dahlberg and Simon, 1969*; *Goff et al., 2012*; *Kingsbury and Darlington, 1968*), hemagglutinating virus of Japan (HVJ) (*Hosaka et al., 1966*), Sendai virus (*Loney et al., 2009*; *Lynch and Kolakofsky, 1978*), measles virus (*Liljeroos et al., 2011*; *Rager et al., 2002*), RSV (*Kiss et al., 2014*) and ebola virus (*Beniac et al., 2012*; *Booth et al., 2013*). The presence of multiple genomes in a virion is essential for their infections, for example, two types of genomic analyses of MeV infections have provided independent evidence of multi-genome MeV transmission (*Rager et al., 2002*; *Shirogane et al., 2012*). In addition, the multi-genome in one virion seems to be packaged in continuous mode in ebola virus (*Beniac et al., 2012*; *Booth et al., 2013*), and even in a 'end to end' mode in HVJ (*Hosaka et al., 1966*). One interesting aspect on self-capping assemblies is that they provide the possibility of accommodating two copies of viral genomes with different lengths in one virion. So, the double-headed mode provides a possible organizing pattern for the multiple genomes of the polyploidy viruses.

In addition, the C-tail of N may exist inside rather than outside of the nucleocapsids of *Mononegavirales*. Even though full-length N was purified and used for structural analysis, the C-tail (residues 399–489) is not easily recognized in the EM map because of the long, intrinsically flexible domain reported in other structures (*Houben et al., 2007*; *Longhi et al., 2003*). Compared to ring structures, the clam-shaped structure has extra cone-like densities in the center (117 nm$^3$ at the threshold of 0.0054) of the density map, which are apparently from the C-tail (*Figure 4—figure supplement 2*). The C-tail is located inside the clam-shaped structure, and is only accessible from either end by P or other proteins in order to form the N–RNA–RdRp complex for the replication and transcription of the genomic RNA; this finding differs from those of previous reports that have described outside-orientated C-tails (*Jensen et al., 2011*; *Krumm et al., 2013*).

NDV infects many domestic and wild avian species, severely impacting the poultry industries in many countries. The structure of NDV N significantly improves our understanding of how NDV protects itself and infects hosts. It is important to highlight that NDV shares many features with other members of the order *Mononegavirales*. For example, measles virus nucleocapsid was reported to assemble into herringbone-shaped structures (*Gutsche et al., 2015*), similar to the NDV $N_{Loop}$ single-headed spiral. Therefore, it is reasonable to predict that nucleocapsid of measles virus, as well as that of other mononegaviruses, might adopt a similar self-capping mechanism in order to keep its genome secure. N is a most conserved viral protein and the vital building block for nucleocapsid assembly, which makes it an ideal target for antivirus drug development (*Cox and Plemper, 2016*). The positive charged clefts between the CTD and NTD lobes of N, and especially the interaction loop between the vertically adjacent N in the clam-shaped structure, are the possible druggable sites for further structure-based design of small-molecule drugs. The structural study of the NDV

provides new insights into the negative-sense RNA virus field and represents the starting point for inspiring new antiviral drug design for mononegavirus diseases.

# Materials and methods

## Key resources table

| Reagent type (species) or resource | Designation | Source or reference | Identifiers | Additional information |
|---|---|---|---|---|
| Gene (Newcastle disease virus) | NDV N | Sangon Biotech Company | GenBank ID: HM063424.1 | Synthetic gene |
| Strain, strain background (E. coli) | BL21 (DE3) Star competent cells | Thermo Fisher Scientific | C6010-03 | Cells for protein expression |
| Strain, strain background (Newcastle disease virus) | LaSota | China Veterinary Culture Collection Center | | |
| Cell line (hamster) | BSR-T7/5 | PMID: 9847328 | | Gift from Zhigao Bu's lab from Harbin Veterinary Research Institute, Chinese Academy of Agricultural Sciences |
| Chemical compound, drug | elastase | SIGMA | E8140-1UN | |
| Chemical compound, drug | RNase A | Promega | A7973 | |
| Software, algorithm | RELION 1.4 | PMID: 23000701 | | https://www3.mrc-lmb.cam.ac.uk/relion/index.php?title=Main_Page |
| Software, algorithm | RELION 2.0 | PMID: 27845625 | | https://www3.mrc-lmb.cam.ac.uk/relion/index.php?title=Main_Page |
| Software, algorithm | UCSF Chimera | http://plato.cgl.ucsf.edu/chimera/ | RRID:SCR_004097 | |
| Software, algorithm | Coot | PMID: 20383002 | RRID:SCR_014222 | http://www2.mrc-lmb.cam.ac.uk/personal/pemsley/coot/ |
| Software, algorithm | PHENIX | PMID: 22505256 | RRID:SCR_014224 | https://www.phenix-online.org/ |
| Software, algorithm | ImageJ | http://imagej.nih.gov/ij/ | RRID: SCR_003070 | |
| Other | Crystal structure of the paramyxovirus parainfluenza virus 5 nucleoprotein–RNA complex | PMID: 25831513 | PDB: 4XJN | |

## Plasmid construction

The nucleoprotein (N) gene of the Newcastle disease virus (NDV) (GenBank ID: HM063424.1) was synthesized by the Sangon Biotech Company (China). The gene was cloned into the pMCSG7 vector with a N-terminal 6 × His tag and a C-terminal 8 × His tag (*Stols et al., 2002*). The transition mutation of amino acids 114–120 to Ala and the truncations caused by deleting the N-arm (residues 1–33), C-tail (residues 399–489), C-arm and C-tail (371-489), as well as combinations of the N-arm, C-arm and C-tail (1–33 and 371–489) of N gene, were also cloned into the pMCSG7 vector and designed as $N_{Loop}$, $N_{\Delta N-arm}$, $N_{\Delta C-tail}$, $N_{\Delta C-arm\Delta C-tail}$ and $N_{\Delta N-arm\Delta C-arm\Delta C-tail}$, respectively. All of the recombinant plasmids with target genes were sequenced to verify their sequences.

## Protein expression and purification

NDV N and its derived mutants were expressed in *Escherichia coli* BL21(DE3) cells and purified using tandem affinity and gel filtration columns. In detail, the cells were grown in LB media at 37°C until the OD600 nm reached 0.6. The target protein was induced at 16°C for an extra 20 hr with the final concentration of IPTG (isopropyl-B-D-1-thiogalactopyranoside) at 0.1 mM. The cells were harvested by centrifugation at 4680 g for 20 min to obtain the sediments. The pellets were resuspended in PBS buffer (137 mM NaCl, 2.7 mM KCl, 50 mM $Na_2HPO_4$, and 10 mM $KH_2PO_4$ (pH 7.4)) and disrupted with microfluidizer. Cell debris was removed by centrifugation at 38,900 g for 50 min. The clarified supernatant was loaded onto a 5 ml nickel-nitrilotriacetic acid (Ni-NTA) resin gravity column (Qiagen), which was preequilibrated with PBS buffer. The column was washed with 50 ml PBS buffer containing 20 mM imidazole followed by a 100 mM imidazole wash. Finally, the protein was eluted using PBS buffer containing 500 mM imidazole. The proteins with His-tags were concentrated and loaded onto a Superdex G200 size-exclusion chromatography column (120 ml, GE Healthcare Life Sciences, USA) preequilibrated with TRIS buffer at pH 8.0 (20 mM Tris-HCl, 150 mM NaCl and 2 mM DTT). The target proteins with endogenous RNA were collected for the following experiments.

The samples obtained above were loaded onto the top of a continuous 10% to 30% (w/v) sucrose gradient in the same TRIS buffer and centrifuged for 6 hr at $16 \times 10^4$ g and 4°C with an SW40 rotor (Beckman). The samples were collected by puncturing the tube and dialyzing in the TRIS buffer.

## Negative stain EM

Grids of N or its mutants for negative-stain EM were prepared as described previously (*Ohi et al., 2004*). Specifically, 4 µl of samples (0.15 mg/ml) were applied to glow-discharged EM grids covered by a thin layer of continuous carbon film and stained with 2% (w/v) uranyl acetate. Negatively stained grids were imaged on a Tecnai Spirit 120 microscope (Thermo Fisher Scientific, USA) operating at 120 kV. Images were recorded at a magnification of ×43,000 and a defocus set to −2 µm, using a 4K × 4K scintillator-based charge-coupled device camera (UltraScan 4000, Gatan, USA).

## Cryo-EM data collection

To prevent sample aggregation, the N-RNA sample was diluted to 0.65 mg/ml containing 0.018 mg/ml Qβ virus-like particles. A 4 µl sample was applied to a glow-discharged holey carbon grid (Quantifoil, R1.2/1.3, Ted Pella) with a thin layer of continuous carbon film. The grids were blotted using a Vitrobot Mark IV (Thermo Fisher Scientific, USA) with 5 s blotting time, force level of 2 at 100% humidity and 4°C and then immediately plunged into liquid ethane cooled by liquid nitrogen.

The micrographs of the clam-shaped structure samples were recorded on a 300 kV Titan Krios $G^2$ electron microscope equipped with Cs corrector (Thermo Fisher Scientific, USA) and a K2 Summit direct electron detector (Gatan, USA), which was used in counting mode with a pixel size of 1.35 Å. Each movie was exposed for 7.6 s and dose-fractioned into 38 frames with 0.2 s for each frame, generating a total dose of ~41 $e^-/A^2$ on the samples. Defocus values during data collection varied from −1.5 µm to −3 µm. All the images were collected under the SerialEM automated data collection software package (*Mastronarde, 2005*). The micrographs of the filament samples were collected on a 200 kV Talos F200C electron microscope (Thermo Fisher Scientific, USA) equipped with a DE20 Summit direct electron detector (DE, USA) in counting mode with a pixel size of 1.582 Å. Each movie was exposed for 40 s and contained 32 frames, generating a total dose of ~41$e^-/A^2$ on the samples. Defocus values for the date collection varied from −1.5 µm to −3 µm. All the images were collected by utilizing the SerialEM automated data collection software package (*Mastronarde, 2005*).

## Cryo-EM data processing and 3D reconstruction

A total of 3200 micrographs were used for the clam-shaped structure determination. Before further image processing, the images were aligned and summed with MotionCorr software (*Li et al., 2013*) and the CTF parameters of each image were determined by CTFFIND3 (*Mindell and Grigorieff, 2003*). The single-particle analysis and reconstruction was mainly executed in Relion1.4 (*Scheres, 2012*) and Relion 2.0 (*Kimanius et al., 2016*). First of all, the particles were picked automatically by Gautomatch and bad particles were excluded by manual selection and reference-free two-dimensional (2D) classification, with 167,588 particles selected for further processing. The initial model was produced by EMAN2 using typical 2D classes with different

orientations (*Tang et al., 2007*). The initial model was lowpass-filtered to 60 Å to limit reference bias during three-dimensional (3D) classification and later refinement. No symmetry was applied in the 3D classification process, and one of the three classes with a better structure feature was selected for further 3D auto-refinement. A 3D map with an overall resolution of 6.4 Å was obtained without symmetry by 3D refinement of the cleaned-up 75,290 particles. Then, a soft mask was applied to avoid the influence of the spreading map on the alignment. Meanwhile, the C2 symmetry was also applied and the final resolution was improved to 4.8 Å with the gold-standard Fourier Shell correlation (FSC) 0.143 criteria. The map was filtered and sharpened during a Relion post-processing session and the local resolution was estimated with Resmap (*Kucukelbir et al., 2014*).

Double-headed filaments were divided into two parts for structure determination: helical filaments and clam-shaped junctions. Both helix and joint parts of the filament were picked manually, 2D classified and 3D reconstructed with Relion2.0 (*Kimanius et al., 2016*). For helical reconstruction, 4909 good segments were selected with 75% overlap. The 3D refinement using a cylinder as the initial map yielded a 15 Å-resolution helical map with the helical twist of $-27.30°$ and a helical rise of 4.78 Å. For single-particle reconstruction of the clam-shaped junction, 2608 particles were manually picked, and the same cylinder in helical reconstruction was used as the initial model. 3D refinement without symmetry yielded a structure, which was used as reference for the next refinement with C2 symmetry. All of the reference structures were pre-filtered to 60 Å to avoid reference bias during the 3D reconstruction. The C2 refinement yielded a map at the resolution of 14 Å. Both the helical map and the C2 symmetric map were filtered and b-factor sharpened during a Relion post-processing session.

The direct FFT analysis of a single-headed filament was performed with the EMAN2 software package (*Tang et al., 2007*). In total, 6333 segments of the single-headed filament samples were manually picked and the helical reconstruction was performed in Relion 2.0 (*Kimanius et al., 2016*).

## Model building and validation

The homology model of N and RNA were generated by Modeller (*Eswar et al., 2008*) using the crystal structure of parainfluenza virus 5 (PDB accession number 4XJN) as the template. Then the pseudo-atomic model of N was flexibly docked into the protomer furthest from the seam in the EM density map with Rosetta software (*Das and Baker, 2008*). The extra density excluding N was assigned as RNA enwrapped between NTD and CTD and docked using poly-Uracils due to the uncertain sequence of RNA in Coot (*Emsley et al., 2010*). The model refinement on an N with six Uracils was carried out using secondary structure restraints to maintain proper stereochemistry in Phenix.refine(v1.12) (*Afonine et al., 2012*). The model was further optimized manually for better local density fitting using Coot (*Emsley et al., 2010*). To prevent overfitting, TLS refinement and weight optimization were used to improve overfitting across a wide range of resolutions. Ramachandran outliers were corrected semi-automatically in Coot, and MolProbity statistics were computed to ensure proper stereochemistry. The model of the N was validated by computing a Fourier shell correlation $(FSC)_{slush}$ with the density map. The revised atomic NDV N and poly-Uracils were duplicated and docked as a rigid body to the other protomers using UCSF Chimera software (*Pettersen et al., 2004*).

## Elastase and RNase A enzymatic assay

Elastase and RNase A were selected to test the susceptibility of the $N_{WT}$ and $N_{Loop}$ samples. A mixture of 40 μl Tris buffer at pH 8.0 (20 mM Tris-HCl, 150 mM NaCl and 2 mM DTT) containing $N_{WT}$ or $N_{Loop}$ (0.15 mg/ml) and 0.275 mg/ml of RNase A was incubated at 37°C and sampled after90 s for negative-stain EM. Forty-five images were captured at $\times49,000$ magnification for each grid and the number of either clam-shaped structures or filaments was counted at different digestion timepoints.

$N_{WT}$ or $N_{Loop}$ (0.15 mg/ml) in TRIS buffer was incubated with 0.1 mg/ml chymotrypsin-like elastase at 37 °C and sampled every 30 min for SDS-PAGE analysis.

## Statistical analysis

For double-headed filaments, the distance between the helix tip and the clam-shaped core were measured using ImageJ software. A total of 1371 filaments from 169 raw micrographs were

statistically counted to calculate the averaged length of the filaments and the percentage of filaments with unequal length of single spiral.

In nuclease and elastase cleavage assay, the number of particles of both clam-shaped structures and filaments of $N_{WT}$ and $N_{Loop}$ were counted at different timepoints such as 0 s, 90 s and 180 s. A total of 120 micrographs were statistically counted.

## MALDI-TOF-MS analysis

The samples of $N_{Loop}$ and $N_{WT}$ after chymotrypsin-like elastase digestion were resolved by SDS PAGE. The resulting gel bands were reduced with 10 mM dithiotreitol in 25 mM $NH_4HCO_3$ at 56°C for 60 min and alkylated by 55 mM iodacetamide in 25 mM $NH_4HCO_3$ in the dark for 45 min at room temperature. The gel pieces were washed with 40 µl of 25 mM $NH_4HCO_3$ for 5 min following the addition of 40 µl acetonitrile and then incubated for 15 min. After the gel pieces were dried in Speedvac for 15 min, proteins were digested with trypsin (100 ng for each band) in 25 mM $NH_4HCO_3$ overnight at 37°C. The samples of $N_{Loop}$ and $N_{WT}$ after trypsin treatment were excised for Ultraflextreme matrix-assisted laser desorption ionization time-of-flight/time-of-flight mass spectrometer (MALDI-TOF/TOF-MS) assay. MALDI data processing was performed by the Peptide Mass Fingerprint method (www.matrixscience.com) using the SwissProt database.

## NDV minigenome assay for the assembly mechanism of the N–RNA complex in vivo

NDV minigenome p-LGT and three helper plasmids pCI-N, pCI-P and pCI-L from the NDV strain ZJ1 were constructed by *Zhang et al. (2005)*. BSR-T7/5 cells stably expressing the phage T7 RNA polymerase, which were developed by *Buchholz et al. (1999)*, were donated by Dr. Zhigao Bu (Harbin Veterinary Institute, China). The cells were maintained in DMEM (Gibco) supplemented with 10% fetal calf serum (FCS) and 1 mg/ml G418, as previously reported (*Peeters et al., 2000*).

Different mutant and truncated versions of N were cloned into the pCI-neo plasmid (Promega) and designated the names $N_{Loop}$, $N_{\Delta N-arm}$, $N_{\Delta C-tail}$, $N_{\Delta C-arm\Delta C-tail}$ and $N_{\Delta N-arm\Delta C-arm\Delta C-tail}$, respectively. The co-transfection was performed with minigenome and helper plasmids as reported previously (*Peeters et al., 2000*; *Zhang et al., 2005*). Briefly, the minigenome p-LGT (3 µg), pCI-P (1.5 µg), pCI-L (1.5 µg), with each different N expression plasmid (3 µg), including wild type pCI-N and pCI-N mutants, were cotransfected into BSR-T7/5 cells expressing T7 RNA polymerase. One co-transfection, in which the N expression plasmid was replaced by vector pCI-neo was also conducted as the negative control. The transfection reagent was PolyJet reagent and the transfection procedure was carried out according to the manufacturer's protocol. At 96 hr posttransfection, the GFP fluorescence of different samples was observed by fluorescence microscopy.

## Ribonucleoprotein complex isolation from NDV strain LaSota

NDV strain LaSota was propagated in 9-day-old specific-pathogen-free (SPF) embryonated chicken eggs at 37°C for 96 hr. The infected allantoic fluid was collected and centrifuged at 4320 g for 30 min to remove the cell debris. The supernatants were then subjected to pelleting through a 20% sucrose cushion at 38,900 g for 2 hr at 4°C. The pellets were resuspended in PBS buffer (pH 7.4) in the presence of the EDTA-free protease inhibitor cocktail complete from Roche Diagnostics, and lysed by five cycles of freezing and thawing (in liquid nitrogen and at 37°C, respectively) (*Schoehn et al., 2004*). The NDV lysate was loaded onto the top of a continuous 10% to 30% (w/v) sucrose gradient in PBS buffer (pH 7.4) and centrifuged for 6 hr at $16 \times 10^4$ g and 4°C with the SW40 rotor (Beckman). The samples were collected by puncturing the tube and dialyzing in PBS buffer. The presence of the N-RNA complex was verified by negative-stain EM.

## Data availability

The cryo-EM density map of clam-shaped structure was deposited in the Electron Microscopy Data Bank (EMDB) with the accession number EMD-9793. The atom coordinates of the single N subunit with six uracils were deposited in the Protein Data Bank (PDB) with the PDB ID 6JC3.

## Acknowledgements

This work was supported by the National Nature Science Foundation of China (grants 31330019 (ZJL), 31770948, 31570875 and 81590761 (SO)), the National Key R&D program of China (2017YFA0504800 (QS)), Yunnan Provincial Science and Technology Department Project (2016FC007) and the Pujiang Talent program (17PJ1406700 (QS)). The Cryo-EM work was performed at the Center for Biological Imaging (CBI), Institute of Biophysics, Chinese Academy of Sciences and the EM facility of the National Center for Protein Science Shanghai (NCPSS). We would like to thank Fei Sun, Gang Ji, Xiaojun Huang, Zhenxi Guo, Deyin Fan, Bolin Zhu, and Shuoguo Li from the CBI, Institute of Biophysics, Chinese Academy of Science (CAS) for helping with EM sample preparation and data collection. We are also grateful to Mi Cao, Liangliang Kong and Junrui Li from the EM facility of NCPSS for cryo-EM data collection. We also thank Baidong Hou at the Institute of Biophysics, CAS for sharing Qβ virus-like particles.

## Additional information

### Funding

| Funder | Grant reference number | Author |
|---|---|---|
| National Natural Science Foundation of China | 31330019 | Zhi-Jie Liu |
| National Natural Science Foundation of China | 31770948 | Songying Ouyang |
| National Natural Science Foundation of China | 31570875 | Songying Ouyang |
| National Natural Science Foundation of China | 81590761 | Songying Ouyang |
| Yunnan Provincial Science and Technology Department | 2016FC007 | Zhi-Jie Liu |
| Pujiang Talent Program | 17PJ1406700 | Qing-Tao Shen |
| National Key R&D Programme of China | 2017YFA0504800 | Qing-Tao Shen |

The funders had no role in study design, data collection and interpretation, or the decision to submit the work for publication.

### Author contributions

Xiyong Song, Conceptualization, Software, Formal analysis, Validation, Investigation, Methodology, Writing—original draft, Writing—review and editing; Hong Shan, Software, Formal analysis, Validation, Methodology, Writing—original draft; Yanping Zhu, Conceptualization, Formal analysis, Validation, Investigation, Methodology; Shunlin Hu, Formal analysis, Validation, Investigation, Methodology; Ling Xue, Validation, Investigation, Methodology; Yong Chen, Jian Gu, Investigation; Wei Ding, Software, Formal analysis, Validation; Tongxin Niu, Software; Songying Ouyang, Conceptualization, Formal analysis, Supervision, Funding acquisition, Project administration; Qing-Tao Shen, Software, Formal analysis, Funding acquisition, Validation, Methodology, Writing—original draft; Zhi-Jie Liu, Conceptualization, Resources, Data curation, Software, Formal analysis, Supervision, Funding acquisition, Validation, Methodology, Writing—original draft, Project administration, Writing—review and editing

### Author ORCIDs

Xiyong Song (iD) https://orcid.org/0000-0002-5088-3783
Zhi-Jie Liu (iD) https://orcid.org/0000-0001-7279-2893

### Decision letter and Author response

Decision letter https://doi.org/10.7554/eLife.45057.029
Author response https://doi.org/10.7554/eLife.45057.030

## Additional files

### Supplementary files

• Transparent reporting form
DOI: https://doi.org/10.7554/eLife.45057.019

### Data availability

The cryo-EM density map has been deposited in EMDB with the accession number EMD-9793. The atom coordinates of the structure have been deposited in PDB with the PDB ID 6JC3.

The following datasets were generated:

| Author(s) | Year | Dataset title | Dataset URL | Database and Identifier |
|---|---|---|---|---|
| Xiyong S, Hong S, Yanping Z, Wei D, Songying O, Qing-Tao S, Zhi-Jie L | 2019 | The Cryo-EM structure of nucleoprotein-RNA complex of Newcastle disease virus | http://www.rcsb.org/structure/6JC3 | RCSB Protein Data Bank, 6JC3 |
| Xiyong S, Hong S, Yanping Z, Wei D, Songying O, Qing-Tao S, Zhi-Jie L | 2019 | The Cryo-EM structure of nucleoprotein-RNA complex of Newcastle disease virus | http://www.ebi.ac.uk/pdbe/entry/emdb/EMD-9793 | Electron Microscopy Data Bank, EMD-9793 |

The following previously published datasets were used:

| Author(s) | Year | Dataset title | Dataset URL | Database and Identifier |
|---|---|---|---|---|
| Alayyoubi M, Leser GP, Kors CA, Lamb RA | 2015 | Structure of the parainfluenza virus 5 nucleocapsid-RNA complex: an insight into paramyxovirus polymerase activity | http://www.rcsb.org/structure/4XJN | PDB, 4XJN |
| Cai S, Li J, Wong MT, Jiao P, Fan H, Liu D, Liao M, Jiang J, Shi M, Lam TT, Ren T, Leung FC | 2011 | Genetic characterization and evolutionary analysis of 4 Newcastle disease virus isolate full genomes from waterbirds in South China during 2003-2007 | https://www.ncbi.nlm.nih.gov/nuccore/HM063424.1 | GenBank, HM063424.1 |

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
