## [Decision Letter]

Thank you for submitting your article "Self-capping of nucleoprotein filaments protects Newcastle Disease Virus genome" for consideration by *eLife*. Your article has been reviewed by three peer reviewers, and the evaluation has been overseen by a Reviewing Editor and Detlef Weigel as the Senior Editor. The following individual involved in review of your submission has agreed to reveal their identity: Z Hong Zhou (Reviewer #1).

The reviewers have discussed the reviews with one another and the Reviewing Editor has drafted this decision to help you prepare a revised submission.

Summary:

This manuscript reports the first high-resolution cryoEM structure of the nucleoprotein (N) protein of Newcastle disease virus (NDV), a negative strand RNA virus. They found that recombinantly expressed N protein forms clam-shaped double-headed filaments that could serve as "caps" or nucleation sites on nucleocapsids for viral RNA replication and/or encapsidation. This novel architecture is potentially of great importance for RNA virus replication. Mutagenesis studies are also provided that are interpreted to show the biological importance of this structure for viral replication. It is not clear how specific the mutational alterations are, i.e., whether the alteration specifically affects formation of the clam-shaped structures or alters the protein more generally. Therefore, there was some concern that these functional studies were over-interpreted. There was also a request from all three reviewers for improved images and analysis of the native RNP structures.

Essential revisions:

1) The essential revision is to provide more convincing native RNP images and analysis of those structures.

2) The functional studies are over-interpreted, so those studies should be removed from the manuscript. The least would be to tone down the conclusions because of the reasons outlined below.

The reviewers have made a number of suggestions in their individual reviews that I think will improve the manuscript. Therefore, I am including those important points below:

*Reviewer #1*:

In this paper, the groups led by Liu, Shen and Ouyang reports the first high-resolution cryoEM structure of the nucleoprotein (N) of Newcastle disease virus (NDV). NDV belongs to a group of negative-sense RNA viruses that also include such medically important viruses as Ebola and Measles. The authors found that recombinantly expressed N forms clam-shaped double-headed filaments with a clam-like core at the middle of the filament. Isolates have flexible structures, ranging from long to short filaments and spirals, as well as a single species containing double spirals. They used the sample fraction containing primarily the double spiral core to determine the structure at 4.8Å resolution by single-particle cryoEM. Since NDV N has 40% sequence identity to the known crystal structure Parainfluenza virus 5 (PIV5) (Alayyoubi et al., 2015), atomic model of NDV N was built needing only minor modifications and shares great similarity to known N structures of other negatively strand RNA viruses (e.g., Figure 1—figure supplement 3B showing nearly identical backbone trace). Placement of this atomic model of N to the clam-shaped reconstruction shows two spirals self-capping each other in a back-to-back manner, revealing a loop (residues 114-120) that is not involved in formation the single-headed helix but responsible for the formation of the clam in the double-headed filament. The authors also carried out cryoEM analyses on the filament fraction and obtained low resolution reconstructions (about 15Å) for both the double heads and the helical segment, which are consistent with the single-particle cryoEM reconstruction of the double-headed clam-like core at higher resolution.

The authors supplement the structural analyses with a series of functional studies with constructs harboring wild-type N, loop 114-120-mutated N (named Nloop), and several other truncation mutants. Most significantly, using mini-genome constructs, the authors demonstrated that minigenome replication was dependent on wild-type, presumably clam-shaped, nucleocapsid. Protease treatment experiments revealed different protection patterns between the double-headed and single-headed helices, yielding 40 kDa verses 30 kDa protein fragment. The clam-shaped filaments were also resistant to nuclease treatment while single-headed filaments formed by Nloop mutant were less resistant to such treatment.

Taking these results together, the authors posit a self-capping mechanism by which the clam-shaped structure forms double-headed filaments to protect the NDV viral genome in vivo. Overall, I think the studies are well designed and the authors uncovered a novel architecture of N that has not been described before for any negative-sense RNA viruses. This model has the potential to stimulate further future studies.

*Reviewer #2*:

This study reports the roughly 5Å structure of a "clam shaped" Newcastle disease virus nucleocapsid by cryo-EM. The clam shape derives from the juxtaposition of 2 NP-RNA complexes packing in a "back to back" manner. This arrangement would juxtapose the 5' end of the 2 encapsidated RNAs. The structure reveals that a loop comprising residues 114-120 participates in hydrogen bonding interactions with the equivalent loop in the back to back arrangement of the two N-RNA spirals. Mutation of all of the residues in that loop results in a defect in a cell based minigenome assay of RNA synthesis, and the bacterial expressed mutant appears distinct by negative-stain EM forming more filaments and less rings. Proteolysis and RNase sensitivity assays indicate the loop mutant produces a distinct pattern of cleavage and the RNA is more sensitive to digestion.

The structural observations of this study are new, and differ from prior work on other nonsegmented negative-sense RNA virus nucleocapsids. The significance of the structural observations is not clear from the manuscript. The manuscript also does an inadequate job of connecting the structures with the biology of negative-strand RNA virus nucleocapsids.

1) The conclusion that NDV N assembles into a clam shaped dimer is based on the isolation of 200Å diameter round-shaped particles. The dimensions of those particles are similar to the ring-like N-RNA structures attained for several of the *Mononegavirales*. What fraction of the round-shaped particles are closed discs as seen for other viruses vs the clam shaped structures?

2) The initial clam shaped structure was at 6.4Å resolution with obvious C2 rotational symmetry. Into this structure the authors dock an NP protomer structure. The NP structure was based on homology modeling with that of PIV5 NP which shares 40% sequence identity. Docking of that protomer into the density map shows that the protomers either side of the "seam" have the lowest resolution. This arrangement leads the authors to a model that the junction comprises two NP molecules that are each bound to the 5' end of an RNA. The best evidence for this arrangement is provided by protomers that are, however, furthest away from the seam. What is the evidence that the RNA is fully coated by NP? Could this arrangement reflect interactions of unencapsidated RNA that helps bring the two molecules together? The authors must determine the length of RNAs that are fully coated directly.

3) Have the authors separately fitted the two lobes of NP into their density map? This would be of particular interest around the "seam" where the resolution is lowest.

4) The conclusion that viral genome replication is dependent on a clam-shaped nucleocapsid reflects a significant over-interpretation of the data at hand.

In Figure 3, the authors perform a minigenome assay to determine whether there are functional consequences for RNA synthesis for the loop mutant. The readout of this assay is GFP expression which indirectly reflects transcription of mRNA from the minigenome, which indirectly reflects encapsidation of the RNA. The mutant could however, encapsidate the template perfectly well, but be defective in recruitment of the polymerase to the template, or could alter polymerase processivity – both of which would affect reporter gene activity. However, the authors interpret this to claim that the clam shaped nucleocapsid is required for genome replication.

A second issue relates to the evidence that the mutation of the loop disrupts the production of "clam-shaped" structures. The negative-stain EM of Figure 3E is offered as evidence that the loop mutant can form filaments, but we have no idea whether those structures bind RNA.

A more rigorous analysis of the RNA bound, and dissection of the step at which the NP mutant affects gene expression is needed to support the conclusion that the clam-shape is required for replication.

5) In Figure 4E the authors provide evidence that the NLoop mutant is more sensitive to nuclease attack. It is curious that the extent of NP-RNA disassembly is unaltered at 90s but is affected at 180s. What is the authors explanation for this? Importantly, the values are expressed as% of remaining particles. How where these particles quantified? Were the equivalent number of like particles tested for Nloop vs Nwt? The description in the Materials and methods is inadequate.

*Reviewer #3*:

In this manuscript, authors use cryoEM to characterize the N protein from NDV. negative sense RNA viral nucleoproteins (N or NP) are critical for several different steps in viral replication cycle and the current structure at 4.8 A (nominal resolution) provides structural insights into the role of NP-NP interactions. Structure derived mutations show loss of function.

Overall the study provides important structural insights. That said, there are many questions that are not addressed in the manuscript with regard to biology. The structure is derived from in vitro purified samples. Loss of function is easier to generate, but hard to interpret. Thus, the authors could/should think about how best to position their work in the context of the biological processes that N protein plays roles in. For example, in Figure 3 D vs E, no biochemial difference, some difference in the micrographs and a large different in the minigenome. Why? what are the biological consequences.

[Editors' note: further revisions were requested prior to acceptance, as described below.]

Thank you for resubmitting your work entitled "Self-capping of nucleoprotein filaments protects Newcastle Disease Virus genome" for further consideration at *eLife*. Your revised article has been favorably evaluated by Detlef Weigel (Senior Editor), a Reviewing Editor, and two reviewers.

The manuscript has been improved but there are some remaining issues that need to be addressed before acceptance, as outlined below:

There are a few areas where the interpretation, conclusions and writing can be improved as outlined in reviewer 2's comments. We feel that this will significantly improve the manuscript.

*Reviewer #1:*

I am satisfied by the authors' responses and the revised manuscript. In particularly, the authors have made an effort to isolate native NPs and revised Figure 1—figure supplement 4. It is understandable that such effort is very laborious and not very fruitful, as also documented in the old literature of this and related viruses.

*Reviewer #2:*

This is a revised manuscript that reports the structure of a clam-shaped NDV nucleocapsid. The functional significance of the clam-shape remains to be formally demonstrated, but the structure itself is interesting and suggests a potential – although untested – model for the process of nucleocapsid assembly.

In revising this work the authors provide new images of the native RNPs – which responds to one of the two main criticisms. The authors state "it is extremely time challenging and time consuming to obtain the nucleoprotein with clam-like structure from the Newcastle disease virus because such kind of nucleoprotein is in very low abundance in the virion." This should also further caution the authors to not over interpret the functional relevance of the structure – who focus a good deal of their response and discussion on polyploid virions. This reviewer found the structure provocative and wondered whether the true relevance of the structure relates to "seeding" of the correct assembly of the nucleocapsid during RNA replication with the net result that on occasion 2 copies of the genome end up in the virion.

There remain some serious overstatements of the results in the title and Abstract and some other inaccuracies in the text.

1) The evidence that the self-capping protects the genome from RNAse is weak. As pointed out in the first review a more rigorous analysis of the bound RNA is needed to make such a conclusion. As it stands, the present data do not tell us anything about the bound RNA itself. Absent this knowledge it is a major over-interpretation that the clam protects the RNA from nuclease digestion. The data of Figure 4E do not look at the RNA, they look only at the presence of clam shapes. Thus it is a significant over interpretation of the data.

2) The statement in the Abstract "Uncovering the helical assembly mechanism of the negative-strand RNA virus will help the development of new antiviral therapies" is unclear and is entirely speculation and should be removed. Similarly the last sentence of the Abstract.

3) Throughout the text the authors correctly state that the nucleocapsid is used for replication and transcription. The polymerase displaces the nucleocapsid protein transiently during transcription and replication. Transcription precedes replication and is hence usually referred to first. It is incorrect to state that the RdRP moves across the nucleocapsid for viral translation (Introduction first paragraph).

4) What is "frontal attack" in the final paragraph of the Introduction? The authors should include some explanation.

---

## [Author Response]

Essential revisions:1) The essential revision is to provide more convincing native RNP images and analysis of those structures.

We thank the reviewers and editors for the opportunity to resubmit a revised version of our manuscript by Song et al. entitled “Self-capping of nucleoprotein filaments protects Newcastle Disease Virus genome”. All authors of this manuscript have carefully discussed the questions raised by the reviewers, and we are trying to address all the questions and hope that our revised manuscript will be accepted for publication at *eLife*.

We agree that the most convincing way is to show the clam-like structure of nucleoprotein complex from the native virus. So, we have attempted to obtain more electron microscopic images of the nucleoprotein-RNA complex purified from the newcastle disease virus propagated in 9-day-old embryonated chicken eggs. We have also obtained more negative stain EM images of the nucleoprotein-RNA complex and added them to Figure 1—figure supplement 4. Please note that it is extremely challenging and time consuming to obtain the nucleoprotein with clam-like structure from the newcastle disease virus because such kind of nucleoprotein is in very low abundance in the virion. We should also point out that several previous reports showed that the nucleocapsids in newcastle disease virus are more prone to be disrupted during purification than those of Hemagglutinating virus of Japan (Hosaka et al., J.Mol.Biol., 1968). This fact added extra hurdles to our effort on obtaining enough samples for the cryo-electron microscopy analysis.

In summary, we managed to obtain the negative-stain images of nucleoprotein filaments from the newcastle disease virus and we could identify the self-capping filaments which is in similar shape as that of overexpressed nucleoprotein filaments observed by electron microscopy. The new negative-stain images have been added into Figure 1—figure supplement 4.

2) The functional studies are over-interpreted, so those studies should be removed from the manuscript. The least would be to tone down the conclusions because of the reasons outlined below.

We have modified and toned down the interpretation of the minigenome studies for verifying the function of the clam-like structure in minigenome description and discussion sections of the manuscript.

Reviewer #2:

[…] The structural observations of this study are new, and differ from prior work on other nonsegmented negative-sense RNA virus nucleocapsids. The significance of the structural observations is not clear from the manuscript. The manuscript also does an inadequate job of connecting the structures with the biology of negative-strand RNA virus nucleocapsids.

We have added the importance and implications of our work to further understanding the function of the NP in the first three paragraphs of the Discussion section of the manuscript as suggested.

1) The conclusion that NDV N assembles into a clam shaped dimer is based on the isolation of 200Å diameter round-shaped particles. The dimensions of those particles are similar to the ring-like N-RNA structures attained for several of the Mononegavirales. What fraction of the round-shaped particles are closed discs as seen for other viruses vs the clam shaped structures?

Many crystal structures of nucleoprotein-RNA complex, such as rabies virus, Respiratory Syncytial Virus, Vesicular stomatitis virus and paramyxovirus parainfluenza virus 5 are ring-like structures with around 200 Å diameter. There are both round shaped and filament shaped nucleocapsids in our NDV N samples after the sucrose gradient centrifugation separation. We managed to determine both shaped components. However, the structure of round shaped sample is the clam shaped nucleoprotein-RNA complex, while the structure of the filament sample is the clam shaped structure with the helical nucleoprotein-RNA complex extending out from the clam shaped structure. There is no closed ring like structure observed in our study.

2) The initial clam shaped structure was at 6.4Å resolution with obvious C2 rotational symmetry. Into this structure the authors dock an NP protomer structure. The NP structure was based on homology modeling with that of PIV5 NP which shares 40% sequence identity. Docking of that protomer into the density map shows that the protomers either side of the "seam" have the lowest resolution. This arrangement leads the authors to a model that the junction comprises two NP molecules that are each bound to the 5' end of an RNA. The best evidence for this arrangement is provided by protomers that are, however, furthest away from the seam. What is the evidence that the RNA is fully coated by NP? Could this arrangement reflect interactions of unencapsidated RNA that helps bring the two molecules together? The authors must determine the length of RNAs that are fully coated directly.

The density map of the RNA strand can be recognized and traced clearly in the clam shaped map (Figure 1—figure supplement 3), whose EMDB ID is EMD-9793.

The seam between the two single spirals is about 6 nm and there is no density for the unencapsidated RNA. Thus we are unable to conclude if unencapsidated RNAs bring the two molecules together, unfortunately.

The clam like structures is not homogeneous in length and as result, the density at far end of filament is smeared (so a soft mask was applied to get a high resolution by averting the influence of the smear density), so it is almost impossible to determine the 3’ end of the RNA strand.

3) Have the authors separately fitted the two lobes of NP into their density map? This would be of particular interest around the "seam" where the resolution is lowest.

Yes, we fitted the nucleoprotein subunit structure of the parainfluenza virus 5 (PDB ID 4XJN) to the clam shaped structure of the NDV separately. We can recognize the subunit of the clam shaped structure clearly even at the regions near the “seam” of the nucleoprotein-RNA complex.

4) The conclusion that viral genome replication is dependent on a clam-shaped nucleocapsid reflects a significant over-interpretation of the data at hand.

Agreed, we have modified and toned down the interpretation of the minigenome studies for verifying the function of the clam-like structure in the minigenome description section and discussion section of the manuscript.

In Figure 3, the authors perform a minigenome assay to determine whether there are functional consequences for RNA synthesis for the loop mutant. The readout of this assay is GFP expression which indirectly reflects transcription of mRNA from the minigenome, which indirectly reflects encapsidation of the RNA. The mutant could however, encapsidate the template perfectly well, but be defective in recruitment of the polymerase to the template, or could alter polymerase processivity – both of which would affect reporter gene activity. However, the authors interpret this to claim that the clam shaped nucleocapsid is required for genome replication.

Agreed, we have modified the interpretation of the minigenome studies as suggested.

A second issue relates to the evidence that the mutation of the loop disrupts the production of "clam-shaped" structures. The negative-stain EM of Figure 3E is offered as evidence that the loop mutant can form filaments, but we have no idea whether those structures bind RNA.

The single headed filament with the loop mutant contains RNA because of the A_260_/A_280_ value of which is about 1.05, which is measured by the Nanodrop instrument.

A more rigorous analysis of the RNA bound, and dissection of the step at which the NP mutant affects gene expression is needed to support the conclusion that the clam-shape is required for replication.

Yes, agreed. However, we can’t finish the experiments of dissection of the NP mutants affecting gene expression in cell level within two months, which means that we are not sure if the NP mutant affected replication or transcription, so we toned down the interpretation of the minigenome assay by stating that the NP mutant affected the function of nucleoprotein-RNA complex.

5) In Figure 4E the authors provide evidence that the NLoop mutant is more sensitive to nuclease attack. It is curious that the extent of NP-RNA disassembly is unaltered at 90s but is affected at 180s. What is the authors explanation for this? Importantly, the values are expressed as% of remaining particles. How where these particles quantified? Were the equivalent number of like particles tested for Nloop vs Nwt? The description in the Materials and methods is inadequate.

Both the wild type and N_Loop_ mutant are about 25% remained at the 90s point after RNase digestion, while the N_Loop_ mutant samples are almost 100% disassembled and 25% of wild type samples remained intact, which means that the samples were digested gradually by the RNase and the wild type samples are more stable than N_Loop_ mutant samples

Both the clam shaped particles (rounded shape particles) and the filament samples were counted. we described this in Materials and methods section “In nuclease cleavage assay, the number of particles of both clam-shaped structures and filaments of N_WT_ and N_Loop_ were counted at different timepoints such as 0s, 90s and 180s.”

Reviewer #3:

In this manuscript, authors use cryoEM to characterize the N protein from NDV. negative sense RNA viral nucleoproteins (N or NP) are critical for several different steps in viral replication cycle and the current structure at 4.8 A (nominal resolution) provides structural insights into the role of NP-NP interactions. Structure derived mutations show loss of function.

*Overall the study provides important structural insights. That said, there are many questions that are not addressed in the manuscript with regard to biology. The structure is derived from* in vitro *purified samples. Loss of function is easier to generate, but hard to interpret. Thus, the authors could/should think about how best to position their work in the context of the biological processes that N protein plays roles in. For example, in Figure 3 D vs E, no biochemial difference, some difference in the micrographs and a large different in the minigenome. Why? what are the biological consequences.*

For the Figure 3D and E, though both the N_WT_ and N_Loop_ are eluted similarly in the void volume of the size exclusion chromatography, yet their three dimensional structures of the N_WT_ and N_Loop_ are different. The N_WT_ has clam-like structure while the N_Loop_ shows single-headed spiral as shown in the Cryo-EM images (Figure 2A and Figure 3C). Thus, N_WT_ and N_Loop_ may function differently in the minigenome assay. The N_WT_ succeeds in the transcription and translation for the target GFP gene, while the NLoop fails to transcribe or translate the GFP gene, for reasons, such as the N_Loop_ can’t keep the genome integrity, or the N_Loop_ affects the formation of N-RNA-RdRp complex by RdRp binding to the nucleocapsid to start the expression of target GFP. But the elaborate mechanism of the N_Loop_ above remains to be illuminated. We also added this relative issue in the discussion section of this manuscript.

[Editors' note: further revisions were requested prior to acceptance, as described below.]

Reviewer #2:

This is a revised manuscript that reports the structure of a clam-shaped NDV nucleocapsid. The functional significance of the clam-shape remains to be formally demonstrated, but the structure itself is interesting and suggests a potential – although untested – model for the process of nucleocapsid assembly.In revising this work the authors provide new images of the native RNPs – which responds to one of the two main criticisms. The authors state "it is extremely time challenging and time consuming to obtain the nucleoprotein with clam-like structure from the Newcastle disease virus because such kind of nucleoprotein is in very low abundance in the virion." This should also further caution the authors to not over interpret the functional relevance of the structure – who focus a good deal of their response and discussion on polyploid virions. This reviewer found the structure provocative and wondered whether the true relevance of the structure relates to "seeding" of the correct assembly of the nucleocapsid during RNA replication with the net result that on occasion 2 copies of the genome end up in the virion.

Agreed, the clam shaped structure of the nucleoprotein-RNA complex has never been reported before, as far as we know, thus we know very little about its functional indications. Due to the limitation of the scope of this manuscript, we have to make extra effort to address the new emerging questions, including the reviewer’s concerns, and hopefully, we can publish the follow up results in the future. However, we’d like to share the current discoveries with the community and speculate the functional influence of the clam shaped structure of the nucleoprotein.

There remain some serious overstatements of the results in the title and Abstract and some other inaccuracies in the text.1) The evidence that the self-capping protects the genome from RNAse is weak. As pointed out in the first review a more rigorous analysis of the bound RNA is needed to make such a conclusion. As it stands, the present data do not tell us anything about the bound RNA itself. Absent this knowledge it is a major over-interpretation that the clam protects the RNA from nuclease digestion. The data of Figure 4E do not look at the RNA, they look only at the presence of clam shapes. Thus it is a significant over interpretation of the data.

Disagree, Figure 4E shows the obvious difference on RNase A digesting NWT and NLoop at different timepoints, where N_WT_ rather than NLoop protects the clam-shaped structure after the RNase A digestion. Also, as matter of fact, the NWT contained RNA because of the A_260_/A_280_ value was about 0.9, while the NLoop didn’t contain RNA due to its low A_260_/A_280_ value of about 0.6 after RNase A digestion. In order to make it more clear, we added the following sentence "Meanwhile, the N_^WT^_ rather than the N_Loop_ contained RNA with an absorbance of A_260_/A_280_ ∼0.9, while that of NLoop was ∼0.6".

2) The statement in the Abstract "Uncovering the helical assembly mechanism of the negative-strand RNA virus will help the development of new antiviral therapies" is unclear and is entirely speculation and should be removed. Similarly the last sentence of the Abstract.

We deleted the sentence "Uncovering the helical assembly mechanism of the negative-strand RNA virus will help the development of new antiviral therapies" and the last sentence of the Abstract.

3) Throughout the text the authors correctly state that the nucleocapsid is used for replication and transcription. The polymerase displaces the nucleocapsid protein transiently during transcription and replication. Transcription precedes replication and is hence usually referred to first. It is incorrect to state that the RdRP moves across the nucleocapsid for viral translation (Introduction first paragraph).

We deleted the word " translation " in the sentence " …the RdRp moves across the nucleocapsid for viral transcription and translation ".

4) What is "frontal attack" in the final paragraph of the Introduction? The authors should include some explanation.

We changed the "frontal attack" to "digestion by proteases".